# Provably Powerful Graph Networks

**Haggai Maron*   Heli Ben-Hamu*   Hadar Serviansky*   Yaron Lipman**
Weizmann Institute of Science
Rehovot, Israel

## Abstract

Recently, the Weisfeiler-Lehman (WL) graph isomorphism test was used to measure the expressive power of graph neural networks (GNN). It was shown that the popular message passing GNN cannot distinguish between graphs that are indistinguishable by the 1-WL test (Morris et al., 2018; Xu et al., 2019). Unfortunately, many simple instances of graphs are indistinguishable by the 1-WL test.

In search for more expressive graph learning models we build upon the recent $k$-order invariant and equivariant graph neural networks (Maron et al., 2019a,b) and present two results:

First, we show that such $k$-order networks can distinguish between non-isomorphic graphs as good as the $k$-WL tests, which are provably stronger than the 1-WL test for $k > 2$. This makes these models strictly stronger than message passing models. Unfortunately, the higher expressiveness of these models comes with a computational cost of processing high order tensors.

Second, setting our goal at building a provably stronger, *simple* and *scalable* model we show that a reduced 2-order network containing just scaled identity operator, augmented with a single quadratic operation (matrix multiplication) has a provable 3-WL expressive power. Differently put, we suggest a simple model that interleaves applications of standard Multilayer-Perceptron (MLP) applied to the feature dimension and matrix multiplication. We validate this model by presenting state of the art results on popular graph classification and regression tasks. To the best of our knowledge, this is the first practical invariant/equivariant model with guaranteed 3-WL expressiveness, strictly stronger than message passing models.

## 1   Introduction

Graphs are an important data modality which is frequently used in many fields of science and engineering. Among other things, graphs are used to model social networks, chemical compounds, biological structures and high-level image content information. One of the major tasks in graph data analysis is learning from graph data. As classical approaches often use hand-crafted graph features that are not necessarily suitable to all datasets and/or tasks (e.g., Kriege et al. (2019)), a significant research effort in recent years is to develop deep models that are able to learn new graph representations from raw features (e.g., Gori et al. (2005); Duvenaud et al. (2015); Niepert et al. (2016); Kipf and Welling (2016); Veličković et al. (2017); Monti et al. (2017); Hamilton et al. (2017a); Morris et al. (2018); Xu et al. (2019)).

Currently, the most popular methods for deep learning on graphs are *message passing neural networks* in which the node features are propagated through the graph according to its connectivity structure (Gilmer et al., 2017). In a successful attempt to quantify the expressive power of message passing models, Morris et al. (2018); Xu et al. (2019) suggest to compare the model's ability to *distinguish* between two given graphs to that of the hierarchy of the Weisfeiler-Lehman (WL) graph isomorphism

---

tests (Grohe, 2017; Babai, 2016). Remarkably, they show that the class of message passing models has limited expressiveness and is not better than the first WL test (1-WL, a.k.a. color refinement). For example, Figure 1 depicts two graphs (i.e., in blue and in green) that 1-WL cannot distinguish, hence indistinguishable by any message passing algorithm.

The goal of this work is to explore and develop GNN models that possess higher expressiveness while maintaining scalability, as much as possible. We present two main contributions. First, establishing a baseline for expressive GNNs, we prove that the recent $k$-order invariant GNNs (Maron et al., 2019a,b) offer a natural hierarchy of models that are as expressive as the $k$-WL tests, for $k \geq 2$. Second, as $k$-order GNNs are not practical for $k > 2$ we develop a simple, novel GNN model, that incorporates standard MLPs of the feature dimension and a matrix multiplication layer. This model, working only with $k = 2$ tensors (the same dimension as the graph input data), possesses the expressiveness of 3-WL. Since, in the WL hierarchy, 1-WL and 2-WL are equivalent, while 3-WL is strictly stronger, this model is provably more powerful than the message passing models. For example, it can distinguish the two graphs in Figure 1. As far as we know, this model is the first to offer both expressiveness (3-WL) and scalability ($k = 2$).

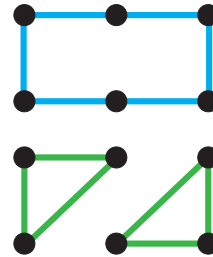

Figure 1: Two graphs not distinguished by 1-WL.

The main challenge in achieving high-order WL expressiveness with GNN models stems from the difficulty to represent the multisets of neighborhoods required for the WL algorithms. We advocate a novel representation of multisets based on Power-sum Multi-symmetric Polynomials (PMP) which are a generalization of the well-known elementary symmetric polynomials. This representation provides a convenient theoretical tool to analyze models' ability to implement the WL tests.

A related work to ours that also tried to build graph learning methods that surpass the 1-WL expressiveness offered by message passing is Morris et al. (2018). They develop powerful deep models generalizing message passing to higher orders that are as expressive as higher order WL tests. Although making progress, their full model is still computationally prohibitive for 3-WL expressiveness and requires a relaxed local version compromising some of the theoretical guarantees.

Experimenting with our model on several real-world datasets that include classification and regression tasks on social networks, molecules, and chemical compounds, we found it to be on par or better than state of the art.

## 2 Previous work

**Deep learning on graph data.** The pioneering works that applied neural networks to graphs are Gori et al. (2005); Scarselli et al. (2009) that learn node representations using recurrent neural networks, which were also used in Li et al. (2015). Following the success of convolutional neural networks (Krizhevsky et al., 2012), many works have tried to generalize the notion of convolution to graphs and build networks that are based on this operation. Bruna et al. (2013) defined graph convolutions as operators that are diagonal in the graph laplacian eigenbasis. This paper resulted in multiple follow up works with more efficient and spatially localized convolutions (Henaff et al., 2015; Defferrard et al., 2016; Kipf and Welling, 2016; Levie et al., 2017). Other works define graph convolutions as local stationary functions that are applied to each node and its neighbours (e.g., Duvenaud et al. (2015); Atwood and Towsley (2016); Niepert et al. (2016); Hamilton et al. (2017b); Veličković et al. (2017); Monti et al. (2018)). Many of these works were shown to be instances of the family of message passing neural networks (Gilmer et al., 2017): methods that apply parametric functions to a node and its neighborhood and then apply some pooling operation in order to generate a new feature for each node. In a recent line of work, it was suggested to define graph neural networks using permutation equivariant operators on tensors describing $k$-order relations between the nodes. Kondor et al. (2018) identified several such linear and quadratic equivariant operators and showed that the resulting network can achieve excellent results on popular graph learning benchmarks. Maron et al. (2019a) provided a full characterization of linear equivariant operators between tensors of arbitrary order. In both cases, the resulting networks were shown to be at least as powerful as message passing neural networks. In another line of work, Murphy et al. (2019) suggest expressive invariant graph models defined using averaging over all permutations of an arbitrary base neural network.

**Weisfeiler Lehman graph isomorphism test.** The Weisfeiler Lehman tests is a hierarchy of increasingly powerful graph isomorphism tests (Grohe, 2017). The WL tests have found many applications in machine learning: in addition to Xu et al. (2019); Morris et al. (2018), this idea was used in Shervashidze et al. (2011) to construct a graph kernel method, which was further generalized to higher order WL tests in Morris et al. (2017). Lei et al. (2017) showed that their suggested GNN has a theoretical connection to the WL test. WL tests were also used in Zhang and Chen (2017) for link prediction tasks. In a concurrent work, Morris and Mutzel (2019) suggest constructing graph features based on an equivalent sparse version of high-order WL achieving great speedup and expressiveness guarantees for sparsely connected graphs.

## 3 Preliminaries

We denote a set by $\{a, b, \ldots, c\}$, an ordered set (tuple) by $(a, b, \ldots, c)$ and a multiset (i.e., a set with possibly repeating elements) by $\{\!\{a, b, \ldots, c\}\!\}$. We denote $[n] = \{1, 2, \ldots, n\}$, and $(a_i \mid i \in [n]) = (a_1, a_2, \ldots, a_n)$. Let $S_n$ denote the permutation group on $n$ elements. We use multi-index $\boldsymbol{i} \in [n]^k$ to denote a $k$-tuple of indices, $\boldsymbol{i} = (i_1, i_2, \ldots, i_k)$. $g \in S_n$ acts on multi-indices $\boldsymbol{i} \in [n]^k$ entrywise by $g(\boldsymbol{i}) = (g(i_1), g(i_2), \ldots, g(i_k))$. $S_n$ acts on $k$-tensors $\mathbf{X} \in \mathbb{R}^{n^k \times a}$ by $(g \cdot \mathbf{X})_{\boldsymbol{i},j} = \mathbf{X}_{g^{-1}(\boldsymbol{i}),j}$, where $\boldsymbol{i} \in [n]^k$, $j \in [a]$.

### 3.1 $k$-order graph networks

Maron et al. (2019a) have suggested a family of permutation-invariant deep neural network models for graphs. Their main idea is to construct networks by concatenating maximally expressive linear equivariant layers. More formally, a $k$-order invariant graph network is a composition $F = m \circ h \circ L_d \circ \sigma \circ \cdots \circ \sigma \circ L_1$, where $L_i : \mathbb{R}^{n^{k_i} \times a_i} \to \mathbb{R}^{n^{k_{i+1}} \times a_{i+1}}$, $\max_{i \in [d+1]} k_i = k$, are *equivariant linear layers*, namely satisfy

$$L_i(g \cdot \mathbf{X}) = g \cdot L_i(\mathbf{X}), \qquad \forall g \in S_n, \quad \forall \mathbf{X} \in \mathbb{R}^{n^{k_i} \times a_i},$$

$\sigma$ is an entrywise non-linear activation, $\sigma(\mathbf{X})_{\boldsymbol{i},j} = \sigma(\mathbf{X}_{\boldsymbol{i},j})$, $h : \mathbb{R}^{n^{k_{d+1}} \times a_{d+1}} \to \mathbb{R}^{a_{d+2}}$ is an *invariant linear layer*, namely satisfies

$$h(g \cdot \mathbf{X}) = h(\mathbf{X}), \qquad \forall g \in S_n, \quad \forall \mathbf{X} \in \mathbb{R}^{n^{k_{d+1}} \times a_{d+1}},$$

and $m$ is a Multilayer Perceptron (MLP). The invariance of $F$ is achieved by construction (by propagating $g$ through the layers using the definitions of equivariance and invariance):

$$F(g \cdot \mathbf{X}) = m(\cdots (L_1(g \cdot \mathbf{X})) \cdots) = m(\cdots (g \cdot L_1(\mathbf{X})) \cdots) = \cdots = m(h(g \cdot L_d(\cdots))) = F(\mathbf{X}).$$

When $k = 2$, Maron et al. (2019a) proved that this construction gives rise to a model that can approximate any message passing neural network (Gilmer et al., 2017) to an arbitrary precision; Maron et al. (2019b) proved these models are universal for a very high tensor order of $k = poly(n)$, which is of little practical value (an alternative proof was recently suggested in Keriven and Peyré (2019)).

### 3.2 The Weisfeiler-Lehman graph isomorphism test

Let $G = (V, E, d)$ be a colored graph where $|V| = n$ and $d : V \to \Sigma$ defines the color attached to each vertex in $V$, $\Sigma$ is a set of colors. The Weisfeiler-Lehman (WL) test is a family of algorithms used to test graph isomorphism. Two graphs $G, G'$ are called isomorphic if there exists an edge and color preserving bijection $\phi : V \to V'$.

There are two families of WL algorithms: $k$-WL and $k$-FWL (Folklore WL), both parameterized by $k = 1, 2, \ldots, n$. $k$-WL and $k$-FWL both construct a coloring of $k$-tuples of vertices, that is $c : V^k \to \Sigma$. Testing isomorphism of two graphs $G, G'$ is then performed by comparing the histograms of colors produced by the $k$-WL (or $k$-FWL) algorithms.

We will represent coloring of $k$-tuples using a tensor $\mathbf{C} \in \Sigma^{n^k}$, where $\mathbf{C}_{\boldsymbol{i}} \in \Sigma$, $\boldsymbol{i} \in [n]^k$ denotes the color of the $k$-tuple $v_{\boldsymbol{i}} = (v_{i_1}, \ldots, v_{i_k}) \in V^k$. In both algorithms, the initial coloring $\mathbf{C}^0$ is defined using the *isomorphism type* of each $k$-tuple. That is, two $k$-tuples $\boldsymbol{i}, \boldsymbol{i}'$ have the same isomorphism type (i.e., get the same color, $\mathbf{C}_{\boldsymbol{i}} = \mathbf{C}_{\boldsymbol{i}'}$) if for all $q, r \in [k]$: (i) $v_{i_q} = v_{i_r} \iff v_{i'_q} = v_{i'_r}$; (ii) $d(v_{i_q}) = d(v_{i'_q})$; and (iii) $(v_{i_r}, v_{i_q}) \in E \iff (v_{i'_r}, v_{i'_q}) \in E$. Clearly, if $G, G'$ are two isomorphic graphs then there exists $g \in S_n$ so that $g \cdot \mathbf{C}'^0 = \mathbf{C}^0$.

In the next steps, the algorithms refine the colorings $\mathbf{C}^l$, $l = 1, 2, \ldots$ until the coloring does not change further, that is, the subsets of $k$-tuples with same colors do not get further split to different color groups. It is guaranteed that no more than $l = poly(n)$ iterations are required (Douglas, 2011).

The construction of $\mathbf{C}^l$ from $\mathbf{C}^{l-1}$ differs in the WL and FWL versions. The difference is in how the colors are aggregated from neighboring $k$-tuples. We define two notions of neighborhoods of a $k$-tuple $\boldsymbol{i} \in [n]^k$:

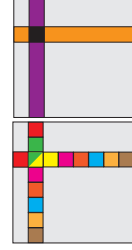

$$N_j(\boldsymbol{i}) = \left\{ (i_1, \ldots, i_{j-1}, i', i_{j+1}, \ldots, i_k) \,\middle|\, i' \in [n] \right\} \tag{1}$$

$$N_j^F(\boldsymbol{i}) = \left( (j, i_2, \ldots, i_k), (i_1, j, \ldots, i_k), \ldots, (i_1, \ldots, i_{k-1}, j) \right) \tag{2}$$

$N_j(\boldsymbol{i})$, $j \in [k]$ is the $j$-th neighborhood of the tuple $\boldsymbol{i}$ used by the WL algorithm, while $N_j^F(\boldsymbol{i})$, $j \in [n]$ is the $j$-th neighborhood used by the FWL algorithm. Note that $N_j(\boldsymbol{i})$ is a set of $n$ $k$-tuples, while $N_j^F(\boldsymbol{i})$ is an ordered set of $k$ $k$-tuples. The inset to the right illustrates these notions of neighborhoods for the case $k = 2$: the top figure shows $N_1(3, 2)$ in purple and $N_2(3, 2)$ in orange. The bottom figure shows $N_j^F(3, 2)$ for all $j = 1, \ldots, n$ with different colors for different $j$.

The coloring update rules are:

$$\text{WL:} \quad \mathbf{C}_{\boldsymbol{i}}^l = \mathrm{enc}\Big( \mathbf{C}_{\boldsymbol{i}}^{l-1}, \Big( \{\!\{ \mathbf{C}_{\boldsymbol{j}}^{l-1} \mid \boldsymbol{j} \in N_j(\boldsymbol{i}) \}\!\} \,\Big|\, j \in [k] \Big) \Big) \tag{3}$$

$$\text{FWL:} \quad \mathbf{C}_{\boldsymbol{i}}^l = \mathrm{enc}\Big( \mathbf{C}_{\boldsymbol{i}}^{l-1}, \{\!\{ ( \mathbf{C}_{\boldsymbol{j}}^{l-1} \mid \boldsymbol{j} \in N_j^F(\boldsymbol{i})) \,\Big|\, j \in [n] \}\!\} \Big) \tag{4}$$

where $\mathrm{enc}$ is a bijective map from the collection of all possible tuples in the r.h.s. of Equations (3)-(4) to $\Sigma$.

When $k = 1$ both rules, (3)-(4), degenerate to $\mathbf{C}_{\boldsymbol{i}}^l = \mathrm{enc}\left( \mathbf{C}_{\boldsymbol{i}}^{l-1}, \{\!\{ \mathbf{C}_{\boldsymbol{j}}^{l-1} \mid j \in [n] \}\!\} \right)$, which will not refine any initial color. Traditionally, the first algorithm in the WL hierarchy is called WL, 1-WL, or the *color refinement algorithm*. In color refinement, one starts with the coloring prescribed with $d$. Then, in each iteration, the color at each vertex is refined by a new color representing its current color and the multiset of its neighbors' colors.

Several known results of WL and FWL algorithms (Cai et al., 1992; Grohe, 2017; Morris et al., 2018; Grohe and Otto, 2015) are:

1. 1-WL and 2-WL have equivalent discrimination power.
2. $k$-FWL is equivalent to $(k+1)$-WL for $k \geq 2$.
3. For each $k \geq 2$ there is a pair of non-isomorphic graphs distinguishable by $(k+1)$-WL but not by $k$-WL.

## 4 Colors and multisets in networks

Before we get to the two main contributions of this paper we address three challenges that arise when analyzing networks' ability to implement WL-like algorithms: (i) Representing the colors $\Sigma$ in the network; (ii) implementing a multiset representation; and (iii) implementing the encoding function.

**Color representation.** We will represent colors as vectors. That is, we will use tensors $\mathbf{C} \in \mathbb{R}^{n^k \times a}$ to encode a color per $k$-tuple; that is, the color of the tuple $\boldsymbol{i} \in [n]^k$ is a vector $\mathbf{C}_{\boldsymbol{i}} \in \mathbb{R}^a$. This effectively replaces the color tensors $\Sigma^{n^k}$ in the WL algorithm with $\mathbb{R}^{n^k \times a}$.

**Multiset representation.** A key technical part of our method is the way we encode multisets in networks. Since colors are represented as vectors in $\mathbb{R}^a$, an $n$-tuple of colors is represented by a matrix $\boldsymbol{X} = [x_1, x_2, \ldots, x_n]^T \in \mathbb{R}^{n \times a}$, where $x_j \in \mathbb{R}^a$, $j \in [n]$ are the rows of $\boldsymbol{X}$. Thinking about $\boldsymbol{X}$ as a multiset forces us to be indifferent to the order of rows. That is, the color representing $g \cdot \boldsymbol{X}$ should be the same as the color representing $\boldsymbol{X}$, for all $g \in S_n$. One possible approach is to perform some sort (e.g., lexicographic) to the rows of $\boldsymbol{X}$. Unfortunately, this seems challenging to implement with equivariant layers.

Instead, we suggest to encode a multiset $\boldsymbol{X}$ using a set of $S_n$-invariant functions called the *Power-sum Multi-symmetric Polynomials* (PMP) (Briand, 2004; Rydh, 2007). The PMP are the multivariate

analog to the more widely known *Power-sum Symmetric Polynomials*, $p_j(y) = \sum_{i=1}^{n} y_i^j$, $j \in [n]$, where $y \in \mathbb{R}^n$. They are defined next. Let $\boldsymbol{\alpha} = (\alpha_1, \ldots, \alpha_a) \in [n]^a$ be a multi-index and for $y \in \mathbb{R}^a$ we set $y^{\boldsymbol{\alpha}} = y_1^{\alpha_1} \cdot y_2^{\alpha_2} \cdots y_a^{\alpha_a}$. Furthermore, $|\boldsymbol{\alpha}| = \sum_{j=1}^{a} \alpha_j$. The PMP of degree $\boldsymbol{\alpha} \in [n]^a$ is

$$p_{\boldsymbol{\alpha}}(\boldsymbol{X}) = \sum_{i=1}^{n} x_i^{\boldsymbol{\alpha}}, \quad \boldsymbol{X} \in \mathbb{R}^{n \times a}.$$

A key property of the PMP is that the finite subset $p_{\boldsymbol{\alpha}}$, for $|\boldsymbol{\alpha}| \leq n$ generates the ring of *Multi-symmetric Polynomials* (MP), the set of polynomials $q$ so that $q(g \cdot \boldsymbol{X}) = q(\boldsymbol{X})$ for all $g \in S_n$, $\boldsymbol{X} \in \mathbb{R}^{n \times a}$ (see, e.g., (Rydh, 2007) corollary 8.4). The PMP generates the ring of MP in the sense that for an arbitrary MP $q$, there exists a polynomial $r$ so that $q(\boldsymbol{X}) = r(u(\boldsymbol{X}))$, where

$$u(\boldsymbol{X}) := \left( p_{\boldsymbol{\alpha}}(\boldsymbol{X}) \mid |\boldsymbol{\alpha}| \leq n \right). \tag{5}$$

As the following proposition shows, a useful consequence of this property is that the vector $u(\boldsymbol{X})$ is a unique representation of the multi-set $\boldsymbol{X} \in \mathbb{R}^{n \times a}$.

**Proposition 1.** *For arbitrary $\boldsymbol{X}, \boldsymbol{X}' \in \mathbb{R}^{n \times a}$: $\exists g \in S_n$ so that $\boldsymbol{X}' = g \cdot \boldsymbol{X}$ if and only if $u(\boldsymbol{X}) = u(\boldsymbol{X}')$.*

We note that Proposition 1 is a generalization of lemma 6 in Zaheer et al. (2017) to the case of multisets of vectors. This generalization was possible since the PMP provide a continuous way to encode *vector* multisets (as opposed to scalar multisets in previous works). The full proof is provided in the supplementary material.

**Encoding function.** One of the benefits in the vector representation of colors is that the encoding function can be implemented as a simple concatenation: Given two color tensors $\mathbf{C} \in \mathbb{R}^{n^k \times a}$, $\mathbf{C}' \in \mathbb{R}^{n^k \times b}$, the tensor that represents for each $k$-tuple $\boldsymbol{i}$ the color pair $(\mathbf{C}_{\boldsymbol{i}}, \mathbf{C}'_{\boldsymbol{i}})$ is simply $(\mathbf{C}, \mathbf{C}') \in \mathbb{R}^{n^k \times (a+b)}$.

## 5  $k$-order graph networks are as powerful as $k$-WL

Our goal in this section is to show that, for every $2 \leq k \leq n$, $k$-order graph networks (Maron et al., 2019a) are at least as powerful as the $k$-WL graph isomorphism test in terms of distinguishing non-isomorphic graphs. This result is shown by constructing a $k$-order network model and learnable weight assignment that implements the $k$-WL test.

To motivate this construction we note that the WL update step, Equation 3, is equivariant (see proof in the supplementary material). Namely, plugging in $g \cdot \mathbf{C}^{l-1}$ the WL update step would yield $g \cdot \mathbf{C}^l$. Therefore, it is plausible to try to implement the WL update step using linear equivariant layers and non-linear pointwise activations.

**Theorem 1.** *Given two graphs $G = (V, E, d)$, $G' = (V', E', d')$ that can be distinguished by the $k$-WL graph isomorphism test, there exists a $k$-order network $F$ so that $F(G) \neq F(G')$. On the other direction for every two isomorphic graphs $G \cong G'$ and $k$-order network $F$, $F(G) = F(G')$.*

The full proof is provided in the supplementary material. Here we outline the basic idea for the proof. First, an input graph $G = (V, E, d)$ is represented using a tensor of the form $\mathbf{B} \in \mathbb{R}^{n^2 \times (e+1)}$, as follows. The last channel of $\mathbf{B}$, namely $\mathbf{B}_{:,:,e+1}$ (':' stands for all possible values $[n]$) encodes the adjacency matrix of $G$ according to $E$. The first $e$ channels $\mathbf{B}_{:,:,1:e}$ are zero outside the diagonal, and $\mathbf{B}_{i,i,1:e} = d(v_i) \in \mathbb{R}^e$ is the color of vertex $v_i \in V$.

Now, the second statement in Theorem 1 is clear since two isomorphic graphs $G, G'$ will have tensor representations satisfying $\mathbf{B}' = g \cdot \mathbf{B}$ and therefore, as explained in Section 3.1, $F(\mathbf{B}) = F(\mathbf{B}')$.

More challenging is showing the other direction, namely that for non-isomorphic graphs $G, G'$ that can be distinguished by the $k$-WL test, there exists a $k$-network distinguishing $G$ and $G'$. The key idea is to show that a $k$-order network can encode the multisets $\{\mathbf{B}_{\boldsymbol{j}} \mid \boldsymbol{j} \in N_j(\boldsymbol{i})\}$ for a given tensor $\mathbf{B} \in \mathbb{R}^{n^k \times a}$. These multisets are the only non-trivial component in the WL update rule, Equation 3. Note that the rows of the matrix $\boldsymbol{X} = \mathbf{B}_{i_1, \ldots, i_{j-1}, :, i_{j+1}, \ldots, i_k, :} \in \mathbb{R}^{n \times a}$ are the colors (i.e., vectors)

that define the multiset $\{\mathbf{B}_{\boldsymbol{j}} \mid \boldsymbol{j} \in N_j(\boldsymbol{i})\}$. Following our multiset representation (Section 4) we would like the network to compute $u(\boldsymbol{X})$ and plug the result at the $\boldsymbol{i}$-th entry of an output tensor $\mathbf{C}$.

This can be done in two steps: First, applying the polynomial function $\tau : \mathbb{R}^a \to \mathbb{R}^b$, $b = \binom{n+a-1}{a-1}$ entrywise to $\mathbf{B}$, where $\tau$ is defined by $\tau(x) = (x^{\boldsymbol{\alpha}} \mid |\boldsymbol{\alpha}| \leq n)$ (note that $b$ is the number of multi-indices $\boldsymbol{\alpha}$ such that $|\boldsymbol{\alpha}| \leq n$). Denote the output of this step $\mathbf{Y}$. Second, apply a linear equivariant operator summing over the $j$-the coordinate of $\mathbf{Y}$ to get $\mathbf{C}$, that is

$$\mathbf{C}_{\boldsymbol{i},:} := L_j(\mathbf{Y})_{\boldsymbol{i},:} = \sum_{i'=1}^{n} \mathbf{Y}_{i_1,\cdots,i_{j-1},i',i_{j+1},\ldots,i_k,:} = \sum_{\boldsymbol{j} \in N_j(\boldsymbol{i})} \tau(\mathbf{B}_{\boldsymbol{j},:}) = u(\boldsymbol{X}), \quad \boldsymbol{i} \in [n]^k,$$

where $\boldsymbol{X} = \mathbf{B}_{i_1,\ldots,i_{j-1},:,i_{j+1},\ldots,i_k,:}$ as desired. Lastly, we use the universal approximation theorem (Cybenko, 1989; Hornik, 1991) to replace the polynomial function $\tau$ with an approximating MLP $m : \mathbb{R}^a \to \mathbb{R}^b$ to get a $k$-order network (details are in the supplementary material). Applying $m$ feature-wise, that is $m(\mathbf{B})_{\boldsymbol{i},:} = m(\mathbf{B}_{\boldsymbol{i},:})$, is in particular a $k$-order network in the sense of Section 3.1.

# 6 A simple network with 3-WL discrimination power

In this section we describe a simple GNN model that has 3-WL discrimination power. The model has the form

$$F = m \circ h \circ B_d \circ B_{d-1} \cdots \circ B_1, \tag{6}$$

where as in $k$-order networks (see Section 3.1) $h$ is an invariant layer and $m$ is an MLP. $B_1, \ldots, B_d$ are blocks with the following structure (see figure 2 for an illustration). Let $\mathbf{X} \in \mathbb{R}^{n \times n \times a}$ denote the input tensor to the block. First, we apply three MLPs $m_1, m_2 : \mathbb{R}^a \to \mathbb{R}^b$, $m_3 : \mathbb{R}^a \to \mathbb{R}^{b'}$ to the input tensor, $m_l(\mathbf{X})$, $l \in [3]$. This means applying the MLP to each feature of the input tensor independently, i.e.,

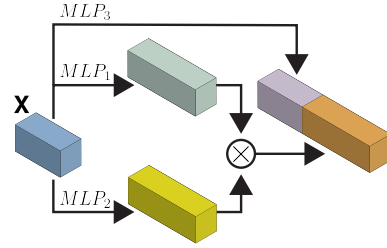

Figure 2: Block structure.

$m_l(\mathbf{X})_{i_1,i_2,:} := m_l(\mathbf{X}_{i_1,i_2,:})$, $l \in [3]$. Second, matrix multiplication is performed between matching features, i.e., $\mathbf{W}_{:,:,j} := m_1(\mathbf{X})_{:,:,j} \cdot m_2(\mathbf{X})_{:,:,j}$, $j \in [b]$. The output of the block is the tensor $(m_3(\mathbf{X}), \mathbf{W})$.

We start with showing our basic requirement from GNN, namely invariance:

**Lemma 1.** *The model $F$ described above is invariant, i.e., $F(g \cdot \mathbf{B}) = F(\mathbf{B})$, for all $g \in S_n$, and $\mathbf{B}$.*

*Proof.* Note that matrix multiplication is equivariant: for two matrices $\boldsymbol{A}, \boldsymbol{B} \in \mathbb{R}^{n \times n}$ and $g \in S_n$ one has $(g \cdot \boldsymbol{A}) \cdot (g \cdot \boldsymbol{B}) = g \cdot (\boldsymbol{A} \cdot \boldsymbol{B})$. This makes the basic building block $B_i$ equivariant, and consequently the model $F$ invariant, i.e., $F(g \cdot \mathbf{B}) = F(\mathbf{B})$. $\square$

Before we prove the 3-WL power for this model, let us provide some intuition as to why matrix multiplication improves expressiveness. Let us show matrix multiplication allows this model to distinguish between the two graphs in Figure 1, which are 1-WL indistinguishable. The input tensor $\mathbf{B}$ representing a graph $G$ holds the adjacency matrix at the last channel $\boldsymbol{A} := \mathbf{B}_{:,:,e+1}$. We can build a network with 2 blocks computing $\boldsymbol{A}^3$ and then take the trace of this matrix (using the invariant layer $h$). Remember that the $d$-th power of the adjacency matrix computes the number of $d$-paths between vertices; in particular $\mathrm{tr}(\boldsymbol{A}^3)$ computes the number of cycles of length 3. Counting shows the upper graph in Figure 1 has 0 such cycles while the bottom graph has 12. The main result of this section is:

**Theorem 2.** *Given two graphs $G = (V, E, d)$, $G' = (V', E', d')$ that can be distinguished by the 3-WL graph isomorphism test, there exists a network $F$ (equation 6) so that $F(G) \neq F(G')$. On the other direction for every two isomorphic graphs $G \cong G'$ and $F$ (Equation 6), $F(G) = F(G')$.*

The full proof is provided in the supplementary material. Here we outline the main idea of the proof. The second part of this theorem is already shown in Lemma 1. To prove the first part, namely that the model in Equation 6 has 3-WL expressiveness, we show it can implement the 2-FWL algorithm, that is known to be equivalent to 3-WL (see Section 3.2). As before, the challenge is in implementing the neighborhood multisets as used in the 2-FWL algorithm. That is, given an input tensor $\mathbf{B} \in \mathbb{R}^{n^2 \times a}$ we would like to compute an output tensor $\mathbf{C} \in \mathbb{R}^{n^2 \times b}$ where $\mathbf{C}_{i_1,i_2,:} \in \mathbb{R}^b$ represents a color matching

the multiset $\big\{\!\!\big\{ (\mathbf{B}_{j,i_2,:}, \mathbf{B}_{i_1,j,:}) \mid j \in [n] \big\}\!\!\big\}$. As before, we use the multiset representation introduced in section 4. Consider the matrix $\boldsymbol{X} \in \mathbb{R}^{n \times 2a}$ defined by

$$\boldsymbol{X}_{j,:} = (\mathbf{B}_{j,i_2,:}, \mathbf{B}_{i_1,j,:}), \quad j \in [n]. \tag{7}$$

Our goal is to compute an output tensor $\mathbf{W} \in \mathbb{R}^{n^2 \times b}$, where $\mathbf{W}_{i_1,i_2,:} = u(\boldsymbol{X})$.

Consider the multi-index set $\big\{ \boldsymbol{\alpha} \mid \boldsymbol{\alpha} \in [n]^{2a}, |\boldsymbol{\alpha}| \leq n \big\}$ of cardinality $b = \binom{n+2a-1}{2a-1}$, and write it in the form $\{ (\boldsymbol{\beta}_l, \boldsymbol{\gamma}_l) \mid \boldsymbol{\beta}, \boldsymbol{\gamma} \in [n]^a, |\boldsymbol{\beta}_l| + |\boldsymbol{\gamma}_l| \leq n, l \in b \}$.

Now define polynomial maps $\tau_1, \tau_2 : \mathbb{R}^a \to \mathbb{R}^b$ by $\tau_1(x) = (x^{\boldsymbol{\beta}_l} \mid l \in [b])$, and $\tau_2(x) = (x^{\boldsymbol{\gamma}_l} \mid l \in [b])$. We apply $\tau_1$ to the features of $\mathbf{B}$, namely $\mathbf{Y}_{i_1,i_2,l} := \tau_1(\mathbf{B})_{i_1,i_2,l} = (\mathbf{B}_{i_1,i_2,:})^{\boldsymbol{\beta}_l}$; similarly, $\mathbf{Z}_{i_1,i_2,l} := \tau_2(\mathbf{B})_{i_1,i_2,l} = (\mathbf{B}_{i_1,i_2,:})^{\boldsymbol{\gamma}_l}$. Now,

$$\mathbf{W}_{i_1,i_2,l} := (\mathbf{Z}_{:,:,l} \cdot \mathbf{Y}_{:,:,l})_{i_1,i_2} = \sum_{j=1}^n \mathbf{Z}_{i_1,j,l} \mathbf{Y}_{j,i_2,l} = \sum_{j=1}^n \mathbf{B}_{j,i_2,:}^{\boldsymbol{\beta}_l} \mathbf{B}_{i_1,j,:}^{\boldsymbol{\gamma}_l} = \sum_{j=1}^n (\mathbf{B}_{j,i_2,:}, \mathbf{B}_{i_1,j,:})^{(\boldsymbol{\beta}_l, \boldsymbol{\gamma}_l)},$$

hence $\mathbf{W}_{i_1,i_2,:} = u(\boldsymbol{X})$, where $\boldsymbol{X}$ is defined in Equation 7. To get an implementation with the model in Equation 6 we need to replace $\tau_1, \tau_2$ with MLPs. We use the universal approximation theorem to that end (details are in the supplementary material).

To conclude, each update step of the 2-FWL algorithm is implemented in the form of a block $B_i$ applying $m_1, m_2$ to the input tensor $\mathbf{B}$, followed by matrix multiplication of matching features, $\mathbf{W} = m_1(\mathbf{B}) \cdot m_2(\mathbf{B})$. Since Equation 4 requires pairing the multiset with the input color of each $k$-tuple, we take $m_3$ to be identity and get $(\mathbf{B}, \mathbf{W})$ as the block output.

**Generalization to $k$-FWL.** One possible extension is to add a generalized matrix multiplication to $k$-order networks to make them as expressive as $k$-FWL and hence $(k+1)$-WL. Generalized matrix multiplication is defined as follows. Given $\mathbf{A}^1, \ldots, \mathbf{A}^k \in \mathbb{R}^{n^k}$, then $(\odot_{i=1}^k \mathbf{A}^i)_{\boldsymbol{i}} = \sum_{j=1}^n \mathbf{A}^1_{j,i_2,\ldots,i_k} \mathbf{A}^2_{i_1,j,\ldots,i_k} \cdots \mathbf{A}^k_{i_1,\ldots,i_{k-1},j}$.

**Relation to (Morris et al., 2018).** Our model offers two benefits over the 1-2-3-GNN suggested in the work of Morris et al. (2018), a recently suggested GNN that also surpasses the expressiveness of message passing networks. First, it has lower space complexity (see details below). This allows us to work with a provably 3-WL expressive model while Morris et al. (2018) resorted to a local 3-GNN version, hindering their 3-WL expressive power. Second, from a practical point of view our model is arguably simpler to implement as it only consists of fully connected layers and matrix multiplication (without having to account for all subsets of size 3).

**Complexity analysis of a single block.** Assuming a graph with $n$ nodes, dense edge data and a constant feature depth, the layer proposed in Morris et al. (2018) has $O(n^3)$ space complexity (number of subsets) and $O(n^4)$ time complexity ($O(n^3)$ subsets with $O(n)$ neighbors each). Our layer (block), however, has $O(n^2)$ space complexity as only second order tensors are stored (i.e., linear in the size of the graph data), and time complexity of $O(n^3)$ due to the matrix multiplication. We note that the time complexity of Morris et al. (2018) can probably be improved to $O(n^3)$ while our time complexity can be improved to $O(n^{2 \cdot x})$ due to more advanced matrix multiplication algorithms.

## 7 Experiments

**Implementation details.** We implemented the GNN model as described in Section 6 (see Equation 6) using the TensorFlow framework (Abadi et al., 2016). We used three identical blocks $B_1, B_2, B_3$, where in each block $B_i : \mathbb{R}^{n^2 \times a} \to \mathbb{R}^{n^2 \times b}$ we took $m_3(x) = x$ to be the identity (i.e., $m_3$ acts as a skip connection, similar to its role in the proof of Theorem 2); $m_1, m_2 : \mathbb{R}^a \to \mathbb{R}^b$ are chosen as $d$ layer MLP with hidden layers of $b$ features. After each block $B_i$ we also added a single layer MLP $m_4 : \mathbb{R}^{b+a} \to \mathbb{R}^b$. Note that although this fourth MLP is not described in the model in Section 6 it clearly does not decrease (nor increase) the theoretical expressiveness of the model; we found it efficient for coding as it reduces the parameters of the model. For the first block, $B_1$, $a = e + 1$, where for the other blocks $b = a$. The MLPs are implemented with $1 \times 1$ convolutions.

Table 1: Graph Classification Results on the datasets from Yanardag and Vishwanathan (2015)

| dataset | MUTAG | PTC | PROTEINS | NCI1 | NCI109 | COLLAB | IMDB-B | IMDB-M |
|---|---|---|---|---|---|---|---|---|
| size | 188 | 344 | 1113 | 4110 | 4127 | 5000 | 1000 | 1500 |
| classes | 2 | 2 | 2 | 2 | 2 | 3 | 2 | 3 |
| avg node # | 17.9 | 25.5 | 39.1 | 29.8 | 29.6 | 74.4 | 19.7 | 13 |
| Results | | | | | | | | |
| GK (Shervashidze et al., 2009) | 81.39±1.7 | 55.65±0.5 | 71.39±0.3 | 62.49±0.3 | 62.35±0.3 | NA | NA | NA |
| RW (Vishwanathan et al., 2010) | 79.17±2.1 | 55.91±0.3 | 59.57±0.1 | > 3 days | NA | NA | NA | NA |
| PK (Neumann et al., 2016) | 76±2.7 | 59.5±2.4 | 73.68±0.7 | 82.54±0.5 | NA | NA | NA | NA |
| WL (Shervashidze et al., 2011) | 84.11±1.9 | 57.97±2.5 | 74.68±0.5 | 84.46±0.5 | 85.12±0.3 | NA | NA | NA |
| FGSD (Verma and Zhang, 2017) | 92.12 | 62.80 | 73.42 | 79.80 | 78.84 | 80.02 | 73.62 | 52.41 |
| AWE-DD (Ivanov and Burnaev, 2018) | NA | NA | NA | NA | NA | 73.93 ± 1.9 | 74.45 ± 5.8 | 51.54 ±3.6 |
| AWE-FB (Ivanov and Burnaev, 2018) | 87.87±9.7 | NA | NA | NA | NA | 70.99 ± 1.4 | 73.13 ±3.2 | 51.58 ± 4.6 |
| DGCNN (Zhang et al., 2018) | 85.83±1.7 | 58.59±2.5 | 75.54±0.9 | 74.44±0.5 | NA | 73.76±0.5 | 70.03±0.9 | 47.83±0.9 |
| PSCN (Niepert et al., 2016)(k=10) | 88.95±4.4 | 62.29±5.7 | 75±2.5 | 76.34±1.7 | NA | 72.6±2.2 | 71±2.3 | 45.23±2.8 |
| DCNN (Atwood and Towsley, 2016) | NA | NA | 61.29±1.6 | 56.61± 1.0 | NA | 52.11±0.7 | 49.06±1.4 | 33.49±1.4 |
| ECC (Simonovsky and Komodakis, 2017) | 76.11 | NA | NA | 76.82 | 75.03 | NA | NA | NA |
| DGK (Yanardag and Vishwanathan, 2015) | 87.44±2.7 | 60.08±2.6 | 75.68±0.5 | 80.31±0.5 | 80.32±0.3 | 73.09±0.3 | 66.96±0.6 | 44.55±0.5 |
| DiffPool (Ying et al., 2018) | NA | NA | 78.1 | NA | NA | 75.5 | NA | NA |
| CCN (Kondor et al., 2018) | 91.64±7.2 | 70.62±7.0 | NA | 76.27±4.1 | 75.54±3.4 | NA | NA | NA |
| Invariant Graph Networks (Maron et al., 2019a) | 83.89±12.95 | 58.53±6.86 | 76.58±5.49 | 74.33±2.71 | 72.82±1.45 | 78.36±2.47 | 72.0±5.54 | 48.73±3.41 |
| GIN (Xu et al., 2019) | 89.4±5.6 | 64.6±7.0 | 76.2±2.8 | 82.7±1.7 | NA | 80.2±1.9 | 75.1±5.1 | 52.3±2.8 |
| 1-2-3 GNN (Morris et al., 2018) | 86.1± | 60.9± | 75.5± | 76.2± | NA | NA | 74.2± | 49.5± |
| Ours 1 | 90.55±8.7 | 66.17±6.54 | 77.2±4.73 | 83.19±1.11 | 81.84±1.85 | 80.16±1.11 | 72.6±4.9 | 50±3.15 |
| Ours 2 | 88.88±7.4 | 64.7±7.46 | 76.39±5.03 | 81.21±2.14 | 81.77±1.26 | 81.38±1.42 | 72.2±4.26 | 44.73±7.89 |
| Ours 3 | 89.44±8.05 | 62.94±6.96 | 76.66±5.59 | 80.97±1.91 | 82.23±1.42 | 80.68±1.71 | 73±5.77 | 50.46±3.59 |
| Rank | 3rd | 2nd | 2nd | 2nd | 2nd | 1st | 6th | 5th |

Parameter search was conducted on learning rate and learning rate decay, as detailed below. We have experimented with two network suffixes adopted from previous papers: (i) The suffix used in Maron et al. (2019a) that consists of an invariant max pooling (diagonal and off-diagonal) followed by a three Fully Connected (FC) with hidden units' sizes of $(512, 256, \#classes)$; (ii) the suffix used in Xu et al. (2019) adapted to our network: we apply the invariant max layer from Maron et al. (2019a) to the output of every block followed by a single fully connected layer to #classes. These outputs are then summed together and used as the network output on which the loss function is defined.

Table 2: Regression, the QM9 dataset.

| Target | DTNN | MPNN | 123-gnn | Ours 1 | Ours 2 |
|---|---|---|---|---|---|
| $\mu$ | 0.244 | 0.358 | 0.476 | **0.231** | **0.0934** |
| $\alpha$ | 0.95 | 0.89 | **0.27** | 0.382 | 0.318 |
| $\epsilon_{homo}$ | 0.00388 | 0.00541 | 0.00337 | **0.00276** | **0.00174** |
| $\epsilon_{lumo}$ | 0.00512 | 0.00623 | 0.00351 | **0.00287** | **0.0021** |
| $\Delta_\epsilon$ | 0.0112 | 0.0066 | 0.0048 | **0.00406** | **0.0029** |
| $\langle R^2 \rangle$ | 17 | 28.5 | 22.9 | **16.07** | **3.78** |
| $ZPVE$ | 0.00172 | 0.00216 | **0.00019** | 0.00064 | 0.000399 |
| $U_0$ | - | - | 0.0427 | 0.234 | **0.022** |
| $U$ | - | - | 0.111 | 0.234 | **0.0504** |
| $H$ | - | - | 0.0419 | 0.229 | **0.0294** |
| $G$ | - | - | 0.0469 | 0.238 | **0.024** |
| $C_v$ | 0.27 | 0.42 | **0.0944** | 0.184 | 0.144 |

**Datasets.** We evaluated our network on two different tasks: Graph classification and graph regression. For classification, we tested our method on eight real-world graph datasets from (Yanardag and Vishwanathan, 2015): three datasets consist of social network graphs, and the other five datasets come from bioinformatics and represent chemical compounds or protein structures. Each graph is represented by an adjacency matrix and possibly categorical node features (for the bioinformatics datasets). For the regression task, we conducted an experiment on a standard graph learning benchmark called the QM9 dataset (Ramakrishnan et al., 2014; Wu et al., 2018). It is composed of 134K small organic molecules (sizes vary from 4 to 29 atoms). Each molecule is represented by an adjacency matrix, a distance matrix (between atoms), categorical data on the edges, and node features; the data was obtained from the pytorch-geometric library (Fey and Lenssen, 2019). The task is to predict 12 real valued physical quantities for each molecule.

**Graph classification results.** We follow the standard 10-fold cross validation protocol and splits from Zhang et al. (2018) and report our results according to the protocol described in Xu et al. (2019), namely the best averaged accuracy across the 10-folds. Parameter search was conducted on a fixed random 90%-10% split: learning rate in $\{5 \cdot 10^{-5}, 10^{-4}, 5 \cdot 10^{-4}, 10^{-3}\}$; learning rate decay in $[0.5, 1]$ every 20 epochs. We have tested three architectures: (1) $b = 400$, $d = 2$, and suffix (ii); (2) $b = 400$, $d = 2$, and suffix (i); and (3) $b = 256$, $d = 3$, and suffix (ii). (See above for definitions of $b, d$ and suffix). Table 1 presents a summary of the results (top part - non deep learning methods). The last row presents our ranking compared to all previous methods; note that we have scored in the top 3 methods in 6 out of 8 datasets.

**Graph regression results.** The data is randomly split into 80% train, 10% validation and 10% test. We have conducted the same parameter search as in the previous experiment on the validation set. We have used the network (2) from classification experiment, i.e., $b = 400$, $d = 2$, and suffix (i), with an absolute error loss adapted to the regression task. Test results are according to the best validation error. We have tried two different settings: (1) training a single network to predict all the output quantities together and (2) training a different network for each quantity. Table 2 compares the mean absolute error of our method with three other methods: 123-gnn (Morris et al., 2018) and (Wu et al., 2018); results of all previous work were taken from (Morris et al., 2018). Note that our method achieves the lowest error on 5 out of the 12 quantities when using a single network, and the lowest error on 9 out of the 12 quantities in case each quantity is predicted by an independent network.

**Equivariant layer evaluation.** The model in Section 6 does not incorporate all equivariant linear layers as characterized in (Maron et al., 2019a). It is therefore of interest to compare this model to models richer in linear equivariant layers, as well as a simple MLP baseline (i.e., without matrix multiplication). We performed such an experiment on the NCI1 dataset (Yanardag and Vishwanathan, 2015) comparing: (i) our suggested model, denoted Matrix Product (MP); (ii) matrix product + full linear basis from (Maron et al., 2019a) (MP+LIN); (iii) only full linear basis (LIN); and (iv) MLP applied to the feature dimension.

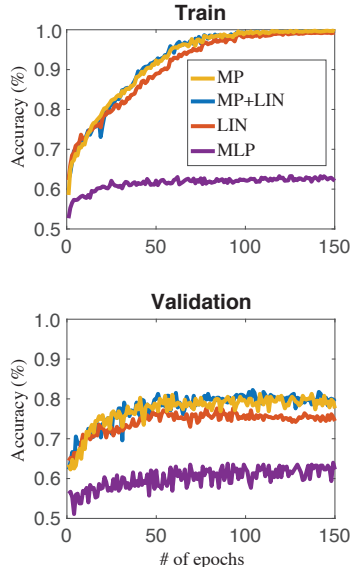

Due to the memory limitation in (Maron et al., 2019a) we used the same feature depths of $b_1 = 32, b_2 = 64, b_3 = 256$, and $d = 2$. The inset shows the performance of all methods on both training and validation sets, where we performed a parameter search on the learning rate (as above) for a fixed decay rate of 0.75 every 20 epochs. Although all methods (excluding MLP) are able to achieve a zero training error, the (MP) and (MP+LIN) enjoy better generalization than the linear basis of Maron et al. (2019a). Note that (MP) and (MP+LIN) are comparable, however (MP) is considerably more efficient.

# 8 Conclusions

We explored two models for graph neural networks that possess superior graph distinction abilities compared to existing models. First, we proved that $k$-order invariant networks offer a hierarchy of neural networks that parallels the distinction power of the $k$-WL tests. This model has lesser practical interest due to the high dimensional tensors it uses. Second, we suggested a simple GNN model consisting of only MLPs augmented with matrix multiplication and proved it achieves 3-WL expressiveness. This model operates on input tensors of size $n^2$ and therefore useful for problems with dense edge data. The downside is that its complexity is still quadratic, worse than message passing type methods. An interesting future work is to search for more efficient GNN models with high expressiveness. Another interesting research venue is quantifying the generalization ability of these models.

**Acknowledgments**

This research was supported in part by the European Research Council (ERC Consolidator Grant, "LiftMatch" 771136) and the Israel Science Foundation (Grant No. 1830/17).

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
