[Supplementary Material]

# Provably Powerful Graph Networks: Supplementary Material

**Haggai Maron**[*]   **Heli Ben-Hamu**[*]   **Hadar Serviansky**[*]   **Yaron Lipman**
Weizmann Institute of Science
Rehovot, Israel

## 1   Proof of Proposition 1

*Proof.* First, if $\boldsymbol{X}' = g \cdot \boldsymbol{X}$, then $p_{\boldsymbol{\alpha}}(\boldsymbol{X}) = p_{\boldsymbol{\alpha}}(\boldsymbol{X}')$ for all $\boldsymbol{\alpha}$ and therefore $u(\boldsymbol{X}) = u(\boldsymbol{X}')$. In the other direction assume by way of contradiction that $u(\boldsymbol{X}) = u(\boldsymbol{X}')$ and $g \cdot \boldsymbol{X} \neq \boldsymbol{X}'$, for all $g \in S_n$. That is, $\boldsymbol{X}$ and $\boldsymbol{X}'$ represent different multisets. Let $[\boldsymbol{X}] = \{g \cdot \boldsymbol{X} \mid g \in S_n\}$ denote the orbit of $\boldsymbol{X}$ under the action of $S_n$; similarly denote $[\boldsymbol{X}']$. Let $K \subset \mathbb{R}^{n \times a}$ be a compact set containing $[\boldsymbol{X}], [\boldsymbol{X}']$, where $[\boldsymbol{X}] \cap [\boldsymbol{X}'] = \emptyset$ by assumption.

By the Stone–Weierstrass Theorem applied to the algebra of continuous functions $C(K, \mathbb{R})$ there exists a polynomial $f$ so that $f|_{[\boldsymbol{X}]} \geq 1$ and $f|_{[\boldsymbol{X}']} \leq 0$. Consider the polynomial

$$q(\boldsymbol{X}) = \frac{1}{n!} \sum_{g \in S_n} f(g \cdot \boldsymbol{X}).$$

By construction $q(g \cdot \boldsymbol{X}) = q(\boldsymbol{X})$, for all $g \in S_n$. Therefore $q$ is a multi-symmetric polynomial. Therefore, $q(\boldsymbol{X}) = r(u(\boldsymbol{X}))$ for some polynomial $r$. On the other hand,

$$1 \leq q(\boldsymbol{X}) = r(u(\boldsymbol{X})) = r(u(\boldsymbol{X}')) = q(\boldsymbol{X}') \leq 0,$$

where we used the assumption that $u(\boldsymbol{X}) = u(\boldsymbol{X}')$. We arrive at a contradiction. □

## 2   Proof of equivairance of WL update step

Consider the formal tensor $\mathbf{B}^j$ of dimension $n^k$ with multisets as entries:

$$\mathbf{B}_{\boldsymbol{i}}^j = \{\mathbf{C}_{\boldsymbol{j}}^{l-1} \mid \boldsymbol{j} \in N_j(\boldsymbol{i})\}. \tag{1}$$

Then the $k$-WL update step (Equation 3) can be written as

$$\mathbf{C}_{\boldsymbol{i}}^l = \mathrm{enc}\left(\mathbf{C}_{\boldsymbol{i}}^{l-1}, \mathbf{B}_{\boldsymbol{i}}^1, \mathbf{B}_{\boldsymbol{i}}^2, \ldots, \mathbf{B}_{\boldsymbol{i}}^k\right). \tag{2}$$

To show equivariance, it is enough to show that each entry of the r.h.s. tuple is equivariant. For its first entry: $(g \cdot \mathbf{C}^{l-1})_{\boldsymbol{i}} = \mathbf{C}_{g^{-1}(\boldsymbol{i})}^{l-1}$. For the other entries, consider w.l.o.g. $\mathbf{B}_{\boldsymbol{i}}^j$:

$$\{(g \cdot \mathbf{C}^{l-1})_{\boldsymbol{j}} \mid \boldsymbol{j} \in N_j(\boldsymbol{i})\} = \{\mathbf{C}_{g^{-1}(\boldsymbol{j})}^{l-1} \mid \boldsymbol{j} \in N_j(\boldsymbol{i})\} = \{\mathbf{C}_{\boldsymbol{j}}^{l-1} \mid \boldsymbol{j} \in N_j(g^{-1}(\boldsymbol{i}))\} = \mathbf{B}_{g^{-1}(\boldsymbol{i})}^j.$$

We get that feeding $k$-WL update rule with $g \cdot \mathbf{C}^{l-1}$ we get as output $\mathbf{C}_{g^{-1}(\boldsymbol{i})}^l = (g \cdot \mathbf{C}^l)_{\boldsymbol{i}}$.

---

[*]Equal contribution

# 3 Proof of Theorem 1

*Proof.* We will prove a slightly stronger claim: Assume we are given some finite set of graphs. For example, we can think of all combinatorial graphs (i.e., graphs represented by binary adjacency matrices) of $n$ vertices . Our task is to build a $k$-order network $F$ that assigns different output $F(G) \neq F(G')$ whenever $G, G'$ are non-isomorphic graphs distinguishable by the $k$-WL test.

Our construction of $F$ has three main steps. First in Section 3.1 we implement the initialization step. Second, Section 3.2 we implement the coloring update rules of the $k$-WL. Lastly, we implement a histogram calculation providing different features to $k$-WL distinguishable graphs in the collection.

## 3.1 Input and Initialization

**Input.** The input to the network can be seen as a tensor of the form $\mathbf{B} \in \mathbb{R}^{n^2 \times (e+1)}$ encoding an input graph $G = (V, E, d)$, as follows. The last channel of $\mathbf{B}$, namely $\mathbf{B}_{:,:,e+1}$ (':' stands for all possible values $[n]$) encodes the adjacency matrix of $G$ according to $E$. The first $e$ channels $\mathbf{B}_{:,:,1:e}$ are zero outside the diagonal, and $\mathbf{B}_{i,i,1:e} = d(v_i) \in \mathbb{R}^e$ is the color of vertex $v_i \in V$. Our assumption of finite graph collection means the set $\Omega \subset \mathbb{R}^{n^2 \times (e+1)}$ of possible input tensors $\mathbf{B}$ is finite as well. Next we describe the different parts of $k$-WL implementation with $k$-order network. For brevity, we will denote by $\mathbf{B} \in \mathbb{R}^{n^k \times a}$ the input to each part and by $\mathbf{C} \in \mathbb{R}^{n^k \times b}$ the output.

**Initialization.** We start with implementing the initialization of $k$-WL, namely computing a coloring representing the isomorphism type of each $k$-tuple. Our first step is to define a linear equivariant operator that extracts the sub-tensor corresponding to each multi-index $\boldsymbol{i}$: let $L : \mathbb{R}^{n^2 \times (e+1)} \to \mathbb{R}^{n^k \times k^2 \times (e+2)}$ be the linear operator defined by

$$L(\mathbf{X})_{\boldsymbol{i},r,s,w} = \mathbf{X}_{i_r,i_s,w}, \quad w \in [e+1]$$

$$L(\mathbf{X})_{\boldsymbol{i},r,s,e+2} = \begin{cases} 1 & i_r = i_s \\ 0 & \text{otherwise} \end{cases}$$

for $\boldsymbol{i} \in [n]^k, r, s \in [k]$.

$L$ is equivariant with respect to the permutation action. Indeed, for $w \in [e+1]$,

$$(g \cdot L(\mathbf{X}))_{\boldsymbol{i},r,s,w} = L(\mathbf{X})_{g^{-1}(\boldsymbol{i}),r,s,w} = \mathbf{X}_{g^{-1}(i_r),g^{-1}(i_s),w} = (g \cdot \mathbf{X})_{i_r,i_s,w} = L(g \cdot \mathbf{X})_{\boldsymbol{i},r,s,w}.$$

For $w = e + 2$ we have

$$(g \cdot L(\mathbf{X}))_{\boldsymbol{i},r,s,w} = L(\mathbf{X})_{g^{-1}(\boldsymbol{i}),r,s,w} = \begin{cases} 1 & g^{-1}(i_r) = g^{-1}(i_s) \\ 0 & \text{otherwise} \end{cases} = \begin{cases} 1 & i_r = i_s \\ 0 & \text{otherwise} \end{cases} = L(g \cdot \mathbf{X})_{\boldsymbol{i},r,s,w}.$$

Since $L$ is linear and equivariant it can be represented as a single linear layer in a $k$-order network. Note that $L(\mathbf{B})_{\boldsymbol{i},:,:,1:(e+1)}$ contains the sub-tensor of $\mathbf{B}$ defined by the $k$-tuple of vertices $(v_{i_1}, \ldots, v_{i_k})$, and $L(\mathbf{B})_{\boldsymbol{i},:,:,e+2}$ represents the equality pattern of the $k$-tuple $\boldsymbol{i}$, which is equivalent to the equality pattern of the $k$-tuple of vertices $(v_{i_1}, \ldots, v_{i_k})$. Hence, $L(\mathbf{B})_{\boldsymbol{i},:,:,:}$ represents the isomorphism type of the $k$-tuple of vertices $(v_{i_1}, \ldots, v_{i_k})$. The first layer of our construction is therefore $\mathbf{C} = L(\mathbf{B})$.

## 3.2 $k$-WL update step

We next implement Equation 3. We achieve that in 3 steps. As before let $\mathbf{B} \in \mathbb{R}^{n^k \times a}$ be the input tensor to the the current $k$-WL step.

First, apply the polynomial function $\tau : \mathbb{R}^a \to \mathbb{R}^b$, $b = \binom{n+a-1}{a-1}$ entrywise to $\mathbf{B}$, where $\tau$ is defined by $\tau(x) = (x^{\boldsymbol{\alpha}})_{|\boldsymbol{\alpha}| \leq n}$ (note that $b$ is the number of multi-indices $\boldsymbol{\alpha}$ such that $|\boldsymbol{\alpha}| \leq n$). This gives $\mathbf{Y} \in \mathbb{R}^{n^k \times b}$ where $\mathbf{Y}_{\boldsymbol{i},:} = \tau(\mathbf{B}_{\boldsymbol{i},:}) \in \mathbb{R}^b$.

Second, apply the linear operator

$$\mathbf{C}^j_{\boldsymbol{i},r} := L_j(\mathbf{Y})_{\boldsymbol{i},r} = \sum_{i'=1}^{n} \mathbf{Y}_{i_1,\cdots,i_{j-1},i',i_{j+1},\ldots,i_k,r}, \quad \boldsymbol{i} \in [n]^k, r \in [b].$$

$L_j$ is equivariant with respect to the permutation action. Indeed, $L_j(g \cdot \mathbf{Y})_{\boldsymbol{i},r} =$

$$\sum_{i'=1}^{n} (g \cdot \mathbf{Y})_{i_1,\cdots,i_{j-1},i',i_{j+1},\ldots,r} = \sum_{i'=1}^{n} \mathbf{Y}_{g^{-1}(i_1)\cdots,g^{-1}(i_{j-1}),i',g^{-1}(i_{j+1}),\ldots,r} = L_j(\mathbf{Y})_{g^{-1}(\boldsymbol{i}),r} = (g \cdot L_j(\mathbf{Y}))_{\boldsymbol{i},r}.$$

Now, note that

$$\mathbf{C}_{\boldsymbol{i},:}^{j} = L_j(\mathbf{Y})_{\boldsymbol{i},:} = \sum_{i'=1}^{n} \tau(\mathbf{B}_{i_1,\cdots,i_{j-1},i',i_{j+1},\ldots,i_k,:}) = \sum_{\boldsymbol{j} \in N_j(\boldsymbol{i})} \tau(\mathbf{B}_{\boldsymbol{j},:}) = u(\boldsymbol{X}),$$

where $\boldsymbol{X} = \mathbf{B}_{i_1,\ldots,i_{j-1},:,i_{j+1},\ldots,i_k,:}$ as desired.

Third, the $k$-WL update step is the concatenation: $(\mathbf{B}, \mathbf{C}^1, \ldots, \mathbf{C}^k)$.

To finish this part we need to replace the polynomial function $\tau$ with an MLP $m : \mathbb{R}^a \to \mathbb{R}^b$. Since there is a finite set of input tensors $\Omega$, there could be only a finite set $\Upsilon$ of colors in $\mathbb{R}^a$ in the input tensors to every update step. Using MLP universality (Cybenko, 1989; Hornik, 1991) , let $m$ be an MLP so that $\|\tau(x) - m(x)\| < \epsilon$ for all possible colors $x \in \Upsilon$. We choose $\epsilon$ sufficiently small so that for all possible $\boldsymbol{X} = (\mathbf{B}_{\boldsymbol{j}} \mid \boldsymbol{j} \in N_j(\boldsymbol{i})) \in \mathbb{R}^{n \times a}$, $\boldsymbol{i} \in [n]^k$, $j \in [k]$, $v(\boldsymbol{X}) = \sum_{i \in [n]} m(x_i)$ satisfies the same properties as $u(\boldsymbol{X}) = \sum_{i \in [n]} \tau(x_i)$ (see Proposition 1), namely $v(\boldsymbol{X}) = v(\boldsymbol{X}')$ iff $\exists g \in S_n$ so that $\boldsymbol{X}' = g \cdot \boldsymbol{X}$. Note that the 'if' direction is always true by the invariance of the sum operator to permutations of the summands. The 'only if' direction is true for sufficiently small $\epsilon$. Indeed, $\|v(\boldsymbol{X}) - u(\boldsymbol{X})\| \le n \max_{i \in [n]} \|m(x_i) - \tau(x_i)\| \le n\epsilon$, since $x_i \in \Upsilon$. Since this error can be made arbitrary small, $u$ is injective and there is a finite set of possible $\boldsymbol{X}$ then $v$ can be made injective by sufficiently small $\epsilon > 0$.

## 3.3 Histogram computation

So far we have shown we can construct a $k$-order equivariant network $H = L_d \circ \sigma \circ \cdots \circ \sigma \circ L_1$ implementing $d$ steps of the $k$-WL algorithm. We take $d$ sufficiently large to discriminate the graphs in our collection as much as $k$-WL is able to. Now, when feeding an input graph this equivariant network outputs $H(\mathbf{B}) \in \mathbb{R}^{n^k \times a}$ which matches a color $H(\mathbf{B})_{\boldsymbol{i},:}$ (i.e., vector in $\mathbb{R}^a$) to each $k$-tuple $\boldsymbol{i} \in [n]^k$.

To produce the final network we need to calculate a feature vector per graph that represents the histogram of its $k$-tuples' colors $H(\mathbf{B})$. As before, since we have a finite set of graphs, the set of colors in $H(\mathbf{B})$ is finite; let $b$ denote this number of colors. Let $m : \mathbb{R}^a \to \mathbb{R}^b$ be an MLP mapping each color $x \in \mathbb{R}^a$ to the one-hot vector in $\mathbb{R}^b$ representing this color. Applying $m$ entrywise after $H$, namely $m(H(\mathbf{B}))$, followed by the summing invariant operator $h : \mathbb{R}^{n^k \times b} \to \mathbb{R}^b$ defined by $h(\mathbf{Y})_j = \sum_{\boldsymbol{i} \in [n]^k} \mathbf{Y}_{\boldsymbol{i},j}$, $j \in [b]$ provides the desired histogram. Our final $k$-order invariant network is

$$F = h \circ m \circ L_d \circ \sigma \circ \cdots \circ \sigma \circ L_1.$$

$\square$

## 4 Proof of Theorem 2

*Proof.* The second claim is proved in Lemma 1. Next we construct a network as in Equation 6 distinguishing a pair of graphs that are 3-WL distinguishable. As before, we will construct the network distinguishing any finite set of graphs of size $n$. That is, we consider a finite set of input tensors $\Omega \subset \mathbb{R}^{n^2 \times (e+2)}$.

**Input.** We assume our input tensors have the form $\mathbf{B} \in \mathbb{R}^{n^2 \times (e+2)}$. The first $e+1$ channels are as before, namely encode vertex colors (features) and adjacency information. The $e+2$ channel is simply taken to be the identity matrix, that is $\mathbf{B}_{:,:,e+2} = I_d$.

**Initialization.** First, we need to implement the 2-FWL initialization (see Section 3.2). Namely, given an input tensor $\mathbf{B} \in \mathbb{R}^{n^2 \times (e+1)}$ construct a tensor that colors 2-tuples according to their

isomorphism type. In this case the isomorphism type is defined by the colors of the two nodes and whether they are connected or not. Let $A := \mathbf{B}_{:,:,e+1}$ denote the adjacency matrix, and $\mathbf{Y} := \mathbf{B}_{:,:,1:e}$ the input vertex colors. Construct the tensor $\mathbf{C} \in \mathbb{R}^{n^2 \times (4e+1)}$ defined by the concatenation of the following colors matrices into one tensor:

$$A \cdot \mathbf{Y}_{:,:,j}, \quad (\mathbf{1}\mathbf{1}^T - A) \cdot \mathbf{Y}_{:,:,j}, \quad \mathbf{Y}_{:,:,j} \cdot A, \quad \mathbf{Y}_{:,:,j} \cdot (\mathbf{1}\mathbf{1}^T - A), \quad j \in [e],$$

and $\mathbf{B}_{:,:,e+2}$. Note that $\mathbf{C}_{i_1,i_2,:}$ encodes the isomorphism type of the 2-tuple sub-graph defined by $v_{i_1}, v_{i_2} \in V$, since each entry of $\mathbf{C}$ holds a concatenation of the node colors times the adjacency matrix of the graph ($A$) and the adjacency matrix of the complement graph ($\mathbf{1}\mathbf{1}^T - A$); the last channel also contains an indicator if $v_{i_1} = v_{i_2}$. Note that the transformation $\mathbf{B} \mapsto \mathbf{C}$ can be implemented with a single block $B_1$.

**2-FWL update step.** Next we implement a 2-FWL update step, Equation 4, which for $k = 2$ takes the form $\mathbf{C}_{\boldsymbol{i}} = \mathrm{enc}\Big(\mathbf{B}_{\boldsymbol{i}}, \big\{\!\!\big\{ (\mathbf{B}_{j,i_2}, \mathbf{B}_{i_1,j}) \,\big|\, j \in [n] \big\}\!\!\big\} \Big)$, $\boldsymbol{i} = (i_1, i_2)$, and the input tensor $\mathbf{B} \in \mathbb{R}^{n^2 \times a}$. To implement this we will need to compute a tensor $\mathbf{Y}$, where the coloring $\mathbf{Y}_{\boldsymbol{i}}$ encodes the multiset $\big\{\!\!\big\{ (\mathbf{B}_{j,i_2,:}, \mathbf{B}_{i_1,j,:}) \,\big|\, j \in [n] \big\}\!\!\big\}$.

As done before, we use the multiset representation described in section 4. Consider the matrix $\boldsymbol{X} \in \mathbb{R}^{n \times 2a}$ defined by

$$\boldsymbol{X}_{j,:} = (\mathbf{B}_{j,i_2,:}, \mathbf{B}_{i_1,j,:}), \quad j \in [n]. \tag{3}$$

Our goal is to compute an output tensor $\mathbf{W} \in \mathbb{R}^{n^2 \times b}$, where $\mathbf{W}_{i_1,i_2,:} = u(\boldsymbol{X})$.

Consider the multi-index set $\big\{ \boldsymbol{\alpha} \,\big|\, \boldsymbol{\alpha} \in [n]^{2a}, |\boldsymbol{\alpha}| \le n \big\}$ of cardinality $b = \binom{n+2a-1}{2a-1}$, and write it in the form $\{(\boldsymbol{\beta}_l, \boldsymbol{\gamma}_l) \mid \boldsymbol{\beta}, \boldsymbol{\gamma} \in [n]^a, |\boldsymbol{\beta}_l| + |\boldsymbol{\gamma}_l| \le n, l \in b\}$. Now define polynomial maps $\tau_1, \tau_2 : \mathbb{R}^a \to \mathbb{R}^b$ by $\tau_1(x) = (x^{\boldsymbol{\beta}_l} \mid l \in [b])$, and $\tau_2(x) = (x^{\boldsymbol{\gamma}_l} \mid l \in [b])$. We apply $\tau_1$ to the features of $\mathbf{B}$, namely $\mathbf{Y}_{i_1,i_2,l} := \tau_1(\mathbf{B})_{i_1,i_2,l} = (\mathbf{B}_{i_1,i_2,:})^{\boldsymbol{\beta}_l}$; similarly, $\mathbf{Z}_{i_1,i_2,l} := \tau_2(\mathbf{B})_{i_1,i_2,l} = (\mathbf{B}_{i_1,i_2,:})^{\boldsymbol{\gamma}_l}$. Now,

$$\mathbf{W}_{i_1,i_2,l} := (\mathbf{Z}_{:,:,l} \cdot \mathbf{Y}_{:,:,l})_{i_1,i_2} = \sum_{j=1}^{n} \mathbf{Z}_{i_1,j,l} \mathbf{Y}_{j,i_2,l} = \sum_{j=1}^{n} \tau_1(\mathbf{B})_{j,i_2,l}\, \tau_2(\mathbf{B})_{i_1,j,l}$$

$$= \sum_{j=1}^{n} \mathbf{B}_{j,i_2,:}^{\boldsymbol{\beta}_l} \mathbf{B}_{i_1,j,:}^{\boldsymbol{\gamma}_l} = \sum_{j=1}^{n} (\mathbf{B}_{j,i_2,:}, \mathbf{B}_{i_1,j,:})^{(\boldsymbol{\beta}_l, \boldsymbol{\gamma}_l)},$$

hence $\mathbf{W}_{i_1,i_2,:} = u(\boldsymbol{X})$, where $\boldsymbol{X}$ is defined in Equation 3.

To implement this in the network we need to replace $\tau_i$ with MLPs $m_i$, $i = 1, 2$. That is,

$$\mathbf{W}_{i_1,i_2,l} := \sum_{j=1}^{n} m_1(\mathbf{B})_{j,i_2,l}\, m_2(\mathbf{B})_{i_1,j,l} = v(\boldsymbol{X}), \tag{4}$$

where $\boldsymbol{X} \in \mathbb{R}^{n \times 2a}$ is defined in Equation 3.

As before, since input tensors belong to a finite set $\Omega \subset \mathbb{R}^{n^2 \times (e+1)}$, so are all possible multisets $\boldsymbol{X}$ and all colors, $\Upsilon$, produced by any part of the network. Similarly to the proof of Theorem 1 we can take (using the universal approximation theorem) MLPs $m_1, m_2$ so that $\max_{x \in \Upsilon, i=1,2} \|\tau_i(x) - m_i(x)\| < \epsilon$. We choose $\epsilon$ to be sufficiently small so that the map $v(\boldsymbol{X})$ defined in Equation 4 maintains the injective property of $u$ (see Proposition 1): It discriminates between $\boldsymbol{X}, \boldsymbol{X}'$ not representing the same multiset.

Lastly, note that taking $m_3$ to be the identity transformation and concatenating $(\mathbf{B}, m_1(\mathbf{B}) \cdot m_2(\mathbf{B}))$ concludes the implementation of the 2-FWL update step. The computation of the color histogram can be done as in the proof of Theorem 1. $\qquad\square$