[Reviews · NeurIPS 2019]

Reviewer 1



Summary ======= The paper studies how to implement the well-known k-WL algorithm, which is a well-investigated heuristic for the graph isomorphism problem, in the language of linear algebra. Consequently, the authors introduce so-called $k$-order graph networks, an extension of GNNs, that have the same power as the k-WL. Specifically, they show how express the $3$-WL (or equiv. the $2$-FWL), and then sketch how their techniques can be used to show the general case. The experimental results indicate that the new layer provides some benefits over the well-known baselines as well as the $k$-GNN layer (Morris, 2018). Assessment ========= Interesting work that extend/sheds a different light on the work of Maroon et al 2019, and Morris et al, 2018. I didn't check the proofs in detail, although the results and high-level ideas seem logical and plausible. As the work is similar to the one of Morris et al, 2018, the authors should put more work in making their contribution clear. Discussion of related work is good.

Reviewer 2



This is a nice theoretical paper that clearly states the mentioned contributions. It is proven that a neural networks augmented with matrix multiplication is proved it achieves 3-WL expressiveness. The quantifying of the generalization ability of these models seems to be very important.

Reviewer 3



The paper puts k-order networks introduced previously in relation to the k-WL test for graph isomorphism. The paper is on a high technical level, but clearly written and generally comprehensible. The results are closely related to two recent papers (Morris et al, 2018, Xu et al., 2019), which are prominently cited and discussed in the abstract and introduction. The paper presents a different formalism based in k-order networks and introduces techniques to obtain models that are as expressive as the k-WL test. In this technical part the relation to the two other papers is less clear. For example, multiset representations have also been proposed by Xu et al., 2019 for a similar task. Why is a different PMP based method introduced by the authors? The advantages of the introduced techniques should be stated more clearly. The contribution of section 6 is more clear, where a 3-WL equivalent model is proposed, which emulates the folklore 2-WL. The approach is claimed to be more efficient than the approach of Morris et al. with the same expressive power. The argument is understandable, but a result in terms of computational complexity or experimental running time is missing. This would strengthen the contribution. Minor remarks: * Section 3.1: Graphs and possible attributes should be introduced. * p6, l.205 ".5" should be removed ===== Edit ===== The authors have provided complexity results with their rebuttal. I have increased my score by one point.

[Author Response · NeurIPS 2019]

We thank the reviewers for their comments and suggestions. We will incorporate the suggestions in our revised version.
Below, we address the main concerns raised in the reviews.

**(R1) Contribution with respect to Maron et al. [2019a].**   Our work builds upon Maron et al. [2019a]. and extends
it in two directions: (i) First, we provide a refined analysis showing that $k$-order invariant networks are at-least
as expressive as $k$-WL test. This extends the 2-order result discussed in Maron et al. to any $k$. (ii) We show that
incorporating higher order equivariant layers (e.g., matrix multiplication) can strictly increase the expressive power of
$k$-order networks (instead of using higher order tensors). We have used this insight to devise a very simple invariant
architecture that empirically outperforms the model suggested in Maron et al. [2019a]. and is theoretically strictly more
expressive.

**(R1,R3) Contribution with respect to Morris et al. [2019]; "Why does your variant improves over, e.g., 1-2-3**
**GNN?"**   The work of Morris et al. [2019] was one of our main inspirations. However, our work does offer two
important contributions with respect to this paper. (i) We propose a simple and practical network architecture with
provable 3-WL expressive power (section 6, Figure 2). Our 3-WL construction offers two main benefits over the
1-2-3-GNN model proposed by Morris et al. [2019]: First, our method requires $O(n^2)$ space per layer compared to
$O(n^3)$ required by Morris et al. (see next question for more details). This allowed us to work with a 3-WL expressive
model in practice while Morris et al. resorted to a local 3-GNN version, hindering their 3-WL expressive power. Second,
from a practical point of view our model is arguably simpler to implement as it only consists of fully connected layers
and matrix multiplication (without having to account for all subsets of size 3). (ii) Our first result (section 5), proves that
the $k$-order invariant networks constructed in Maron et al. [2019a] from first principles (i.e., equivariance to permutation
action on graphs) surprisingly lead to neural models that can implement $k$-WL. The $k$-order invariant networks therefore
provide an *alternative* theoretical model to Morris et al. [2019], with provable $k$-WL expressive power. We believe the
community would benefit from exploring different frameworks (although they have the same expressive power). We
will add a discussion in the paper.

**(R3) "Complexity results on the approach with 3-WL expressive power would be desirable."**   We can provide
the following complexity analysis for a single layer (i.e., block, see Figure 2). Note that both our method and Morris et
al. [2019] use only a few (e.g., 3) layers in the experiments. Assuming a graph with $n$ nodes, dense edge data and a
constant feature depth, the layer proposed in Morris et al. has $O(n^3)$ space complexity (number of subsets) and $O(n^4)$
time complexity ($O(n^3)$ subsets with $O(n)$ neighbors each). Our layer (block), however, has $O(n^2)$ space complexity
as only second order tensors are stored (i.e., linear in the size of the graph data), and time complexity of $O(n^3)$ due to
the matrix multiplication. We note that the time complexity of Morris et al. can probably be improved to $O(n^3)$ while
our time complexity can be improved to $O(n^{2.x})$ due to more advanced matrix multiplication algorithms.

**(R3) Contribution with respect to Xu et al. [2019].**   Xu et al. [2019] show that message passing neural networks
have limited 1-WL expressive power, and offer a message-passing model that realizes this bound. Our paper extends
this result and provides a theoretical connection between a recently suggested method (Maron et al. [2019a]) and
higher-order WL tests. Moreover, we propose an architecture that is provably more expressive than the model suggested
in Xu et al., with 3-WL expressive power, and compares favorably empirically.

**(R1) "Is it really necessary to resort to the "hammer" of the universal approximation theorem?"**   This is an
interesting question. First of all, it is not clear why using the universal approximation theorem is considered to be a
negative thing. Nevertheless, we admit that trying to prove our results without using it is an intriguing question.

**(R1) "Are your results tight in terms of shapes of the maps?"**   We are not sure we understand the question. If the
reviewer asks whether $k$-order networks cannot implement WL test of degree higher than k, then we did not prove
such a result, although it is likely that this statement is correct. One thing that we do discuss at the end of section 6 is
a potential way for k-order networks, augmented with an additional polynomial operation, to be as expressive as the
$k + 1$-WL test.

**(R2) Code.**   We will release code and instructions for reproducing the results in the paper.

**(R3) "Why is a different PMP based method introduced by the authors?"**   The power sum multisymmetric
polynomials (PMP) provide a continuous and differentiable way to encode *vector* multisets. Previous works, e.g.,
Zaheer et al. [2017], used the power-sum symmetric polynomials to encode *scalar* multisets, or a delicate and somewhat
incomplete exponential encoding scheme. Our approach using PMP provides a much simpler analysis.

[Meta-Review · NeurIPS 2019]

Similar to the k-WL graph kernel approach, the authors present $k$-order graph neural networks. The prove connections to the WL algorithm and also show empirical evidence. Overall, the reviews as well as the discussion clearly show that the connection established and the results, both on theory as well as in praxis are interesting and should be published.